# Stacking Ensemble Deep Learning for Real-Time Intrusion Detection in IoMT Environments

**DOI:** 10.3390/s25030624

**Published:** 2025-01-22

**Authors:** Easa Alalwany, Bader Alsharif, Yazeed Alotaibi, Abdullah Alfahaid, Imad Mahgoub, Mohammad Ilyas

**Affiliations:** 1College of Computer Science and Engineering, Taibah University, Yanbu 46421, Saudi Arabia; ealwani@taibahu.edu.sa (E.A.); afhed@taibahu.edu.sa (A.A.); 2Department of Electrical Engineering and Computer Science, Florida Atlantic University, 777 Glades Road, Boca Raton, FL 33431, USA; mahgoubi@fau.edu (I.M.); ilyas@fau.edu (M.I.); 3Department of Computer Science and Engineering, College of Telecommunication and Information, Technical and Vocational Training Corporation, Riyadh 12464, Saudi Arabia; 4Ministry of National Guard, King Khalid Military Academy, Riyadh 14625, Saudi Arabia; yhmotaibi@sang.gov.sa

**Keywords:** Internet of Medical Things (IoMT), cybersecurity in healthcare, intrusion detection system (IDS), machine and deep learning in IoMT security, stacking method, Kappa Architecture

## Abstract

The Internet of Medical Things (IoMT) is revolutionizing healthcare by enabling advanced patient care through interconnected medical devices and systems. However, its critical role and sensitive data make it a prime target for cyber threats, requiring the implementation of effective security solutions. This paper presents a novel intrusion detection system (IDS) specifically designed for IoMT networks. The proposed IDS leverages machine learning (ML) and deep learning (DL) techniques, employing a stacking ensemble method to enhance detection accuracy by integrating the strengths of multiple classifiers. To ensure real-time performance, the IDS is implemented within a Kappa Architecture framework, enabling continuous processing of IoMT data streams. The system effectively detects and classifies a wide range of cyberattacks, including ARP spoofing, DoS, Smurf, and Port Scan, achieving an outstanding detection accuracy of 0.991 in binary classification and 0.993 in multi-class classification. This research highlights the potential of combining advanced ML and DL methods with ensemble learning to address the unique cybersecurity challenges of IoMT systems, providing a reliable and scalable solution for safeguarding healthcare services.

## 1. Introduction

IoMT represents a specialized branch of the broader Internet of Things (IoT), uniquely tailored to the healthcare sector by interconnecting medical devices, sensors, and healthcare systems to monitor, diagnose, and treat patients remotely. While IoT encompasses a vast array of connected devices across diverse fields, IoMT specifically focuses on critical health-related applications where data security, accuracy, and reliability are paramount to patient well-being. This distinction places IoMT in a high-stakes category where the protection of data and device integrity directly impacts patient health and safety [1,2].

IDSs have traditionally played a pivotal role in safeguarding IoT networks from unauthorized access and cyberattacks. However, IoMT devices face unique challenges, including stringent regulatory requirements, high sensitivity to disruptions, and diverse deployment environments, spanning from hospitals and clinics to remote home-based care. Additionally, these devices manage vast quantities of sensitive health data, such as patient records, real-time monitoring results, and treatment information, which makes them prime targets for cyber threats. The sensitive data are also governed by strict regulations (such as HIPAA in the United States), mandating a high level of security and privacy. Therefore, the application of IDS in IoMT environments requires a comprehensive, adaptive approach to address the unique characteristics and risks associated with healthcare technology [3,4].

In this context, ML and DL have emerged as powerful tools to enhance IDS capabilities, enabling dynamic, real-time detection of complex, ever-evolving threats. Unlike traditional IDS methods, which may struggle to handle the high volume and complex patterns within IoMT-generated data streams, ML- and DL-based approaches can effectively analyze these intricate data structures [2,5]. By employing sophisticated algorithms, such as neural networks and clustering techniques, ML and DL systems can quickly detect patterns, identify anomalies, and classify potential threats with remarkable precision [6]. This is particularly beneficial in healthcare settings, where disruptions in medical device functionality or security breaches could lead to adverse patient outcomes [2,7,8].

Ensemble learning is a powerful methodology in machine and deep learning that combines the predictions of multiple models to achieve better performance than any individual model. By aggregating different ML and DL models, ensemble methods like stacking, bagging, and boosting can effectively reduce errors, improve generalization, and enhance robustness against overfitting. This approach is particularly crucial when dealing with complex datasets like those generated by IoMT, where the stakes for accuracy and efficiency are high. Ensemble learning amplifies detection capabilities by leveraging the strengths of various classifiers, ensuring more precise, reliable, and dynamic threat detection. For instance, in cybersecurity, ensemble methods can better differentiate between normal and anomalous patterns, reducing false positives and increasing system reliability [2,7,8,9]. The effectiveness of IDSs utilizing AI algorithms in IoMT environments comes from their ability to adapt to dynamic changes and address critical challenges, including heterogeneity, scalability, latency, and resource constraints. As highlighted in [10], IDS solutions focused solely on network traffic monitoring are especially appropriate for resource-limited IoMT devices. When combined with advanced data processing frameworks like the Kappa Architecture, the benefits of ensemble learning can be fully realized in real-time environments, enabling dynamic and continuous threat detection.

The Kappa Architecture is a cutting-edge data processing framework designed to handle continuous, high-speed data streams in real-time, making it highly relevant for IoMT environments. Unlike traditional batch processing systems, Kappa Architecture processes and analyzes data as they arrive, ensuring minimal latency in threat detection and response [11]. This real-time capability is critical for IoMT, where delays in identifying cyber threats could directly impact patient safety. By integrating machine and deep learning-based IDS into the Kappa Architecture, this approach ensures that incoming data streams, such as those from medical sensors or devices, are continuously monitored for anomalies [12,13]. The architecture’s adaptability allows it to handle diverse attack scenarios like ARP spoofing and DoS attacks efficiently, while its scalability ensures it can accommodate the growing data influx typical of IoMT networks. This synergy of real-time processing and advanced analytics establishes a solid defense mechanism, safeguarding healthcare systems against evolving cyber threats [11,12].

Ensuring the security of IoMT systems is essential to protect the functionality of healthcare services, maintain data integrity, and ensure patient safety. Designing an effective security scheme is crucial to protecting these interconnected systems. This study proposes a solution for developing an effective IDS scheme. Our scheme develops IDS using ML and DL models, enhanced by a stacking ensemble method to improve detection accuracy. Implemented within a Kappa Architecture, the IDS achieves real-time performance, which is essential for timely threat detection in IoMT environments. The scheme is evaluated using binary classification to distinguish normal from attack samples and multi-class classification to detect and classify four types of attacks: ARP spoofing, DoS, Smurf, and Port Scan.

Our main contributions to this research paper are as follows:We utilize ML and DL models to develop an IDS that is designed to achieve both real-time performance and high detection accuracy.Our scheme implements a stacking ensemble method to enhance the efficiency of detection attacks by combining predictions from multiple ML and DL classifiers.The IDS is implemented within the Kappa Architecture, which allows for the real-time processing of continuous data streams, making it highly suited for IoMT environmentsThe scheme’s effectiveness in detecting and classifying ARP spoofing, DoS, Smurf, Port Scan attacks, compared to other proposed models using the same dataset, is demonstrated by its performance on well-known metrics such as accuracy, precision, recall, and F1 score on an ECU-IoHT dataset.

The structure of this paper is as follows: Section 2 reviews related work and discusses solutions for securing IoMT. Section 3 details the study’s methodology, outlining the specific approaches and techniques employed. Section 4 presents the findings and examines the application of ensemble learning techniques. Finally, Section 5 concludes the paper.

## 2. Related Work

Ref. [14] focused on the critical vulnerabilities associated with the open architecture of IoMT networks, exposing them to cyber threats that can compromise data integrity, confidentiality, and system availability. These vulnerabilities highlight the need for promising attack detection solutions to protect patient safety and ensure reliable healthcare services. Due to the limitations of IoMT devices, traditional detection methods often fall short, necessitating more accurate solutions. Addressing this need, Ref. [15] explored artificial intelligence (AI) techniques, including ML and DL techniques, to enhance anomaly detection. By integrating data from various sources, these AI-driven methods strengthen security in complex healthcare environments, thereby improving IoMT system robustness. Recent research has highlighted the growing importance of data assimilation approaches and sensor optimization in improving field reconstruction in complex scenarios with sparse and mobile sensors. Efficient deep learning-based models, such as Voronoi tessellation-assisted convolutional neural networks (CNNs), have shown considerable potential in managing time-varying sensors and sparse observations in reactor monitoring systems [16]. In addition, deep data assimilation techniques that combine unstructured sensor data through CNNs have improved the accuracy and robustness of predictions in high-dimensional systems, even with dynamic sensor placements [17].

Albattah et al. [18] presented a hybrid convolutional long short-term memory (ConvLSTM) model designed to detect adversarial attacks in healthcare monitoring systems. Using the PhysioNet dataset, their model effectively identifies anomalies caused by adversarial techniques, including fast gradient sign (FGSM) and basic iterative methods (BIM), across both training and testing stages. Achieving an accuracy of 98% and an F1 score of 97%, this approach demonstrates notable robustness in detecting anomalies within IoMT.

One AI-driven model mentioned in [19] utilizes machine learning algorithms, including decision trees, support vector machines (SVMs), logistic regression, and k-nearest neighbors (KNN), to protect medical data in IoMT environments. Using a heart disease dataset from IEEE Data Port, the study aimed to ensure that only authorized users could access sensitive patient information, effectively defending against man-in-the-middle and replay attacks. Among the models tested, KNN achieved particularly high accuracy. Ravi et al. [20] proposed a deep learning-based IDS specifically designed for IoMT networks, leveraging both CNN and LSTM layers to extract features from network flow and patient biometric data. To address data imbalance, the model incorporates a cost-sensitive learning approach along with a global attention mechanism. Evaluated on the WUSTL-EHMS 2020 dataset, the system demonstrated robust attack detection capabilities, achieving 99% accuracy for combined features. Mosaiyebzadeh et al. [21] proposed a federated learning-based IDS for IoMT environments. Their DNN-FL model enhances privacy and security by leveraging deep neural networks within a federated learning (FL) framework, removing the need to share sensitive data with a centralized server. The model was evaluated using the WUSTL-EHMS 2020 and ECU-IoHT datasets, achieving an accuracy of 91.40% and 98.47%, respectively.

Another noteworthy approach in IoMT security was presented in [22], where an embedded federated learning-driven LSTM (EFL-LSTM) model was developed to detect intrusions within IoHT networks. This model combines LSTM with federated learning to identify anomalies in time series data while preserving data privacy. Using the ECU-IoHT dataset, the study achieved a 97.16% accuracy in classifying intrusions. Additionally, RUSBoost is employed as an ensemble method for feature selection, enhancing model performance on imbalanced datasets.

A neighborhood-component-based LSTM model was proposed by Kumar et al. [23] to improve the detection of cyberattacks in IoHT networks. The study utilized neighborhood component analysis (NCA) to select features from the ECU-IoHT dataset, followed by two LSTM variants: directed acyclic graph LSTM (DAG-LSTM) and projected layer LSTM (PL-LSTM). The DAG-LSTM model effectively classified multiple types of attacks, including DoS and ARP spoofing, achieving an average testing accuracy of 92.04%.

A recent study by [24] proposed a stacked hybrid AI technique to improve IoT network security through anomaly detection. This model combines multiple classifiers, including random forest (RF), decision tree (DT), and Naïve Bayes (NB), into a stacked ensemble system to enhance detection accuracy. The model, tested on the UNSW-NB15 dataset, achieved a notable 94.94% accuracy, demonstrating its capability to identify complex patterns within IoT traffic data. The authors emphasized that this stacked ensemble approach is particularly well-suited for real-time IoT security applications due to its effectiveness in recognizing complex anomaly patterns.

Previous studies, such as Refs. [7,8], have significantly contributed to the advancement of IoT security. However, Ref. [8] relied exclusively on machine learning (ML) techniques, limiting its ability to effectively manage the complex patterns inherent in IoMT traffic. Similarly, Ref. [7] addressed intrusion detection in general IoT environments using ensemble ML techniques but did not adequately address the unique challenges faced in healthcare-specific scenarios. Moreover, neither study incorporated a real-time framework, as both depended on batch processing methodologies that introduced latency. Both approaches are restricted to binary classification, focusing solely on distinguishing between normal and attack traffic. In contrast, this study builds upon and advances prior research by integrating both ML and deep learning (DL) methods. This integration enhances the capability to analyze complex and high-dimensional IoMT data, a critical requirement for healthcare applications. The adoption of the Kappa Architecture further strengthens this study by enabling continuous, real-time data processing, effectively addressing latency issues that are crucial for ensuring patient safety in IoMT environments. Additionally, this study supports both binary and multi-class classification, allowing for the detection and categorization of diverse attack types, including ARP spoofing, DoS, Smurf, and Port Scan.

Table 1 provides a comprehensive comparison of the proposed scheme with recently developed models, evaluating key aspects such as the use of ML, DL, ensemble methods, real-time architectures, binary classification, multi-class classification, and type of dataset being used.

## 3. Methodology

IoMT offers numerous applications that enhance patient care and support physicians. By improving patient comfort, streamlining treatment processes, and creating a more efficient environment, IoMT plays a crucial role in modern healthcare. However, securing IoMT systems is essential to protect patient safety, data integrity, and the overall functionality of healthcare services. Therefore, effective security schemes are necessary to safeguard these interconnected systems and ensure their reliability.

This paper proposes a scheme designed to achieve both high detection accuracy and real-time functionality for IoMT security. The proposed scheme develops an intrusion detection system (IDS) to detect and classify attacks by leveraging machine learning (ML) and deep learning (DL) models. To enhance detection accuracy, a stacking ensemble method is employed. This method integrates predictions from multiple ML and DL models, improving the system’s ability to detect complex threats by capitalizing on the strengths of different classifiers. As a result, the IDS achieves enhanced reliability and precision in threat detection.

The scheme is implemented within a Kappa Architecture framework to ensure real-time performance. This architecture focuses on a streaming layer that updates with new data, eliminating the need for batch processing. The continuous processing of incoming data streams with low latency enables timely threat detection, which is critical for IoMT environments.

To evaluate the effectiveness of the proposed IDS, binary classification is used to distinguish between normal and attack samples, while multi-class classification assesses its ability to detect and classify various attack types. The scheme’s performance is tested on an IoT dataset, with key metrics such as accuracy, precision, recall, and F1 score reported for both scenarios. This comprehensive evaluation highlights the IDS’s capability to operate effectively in real-time IoMT environments.

The workflow of the proposed model, as illustrated in Figure 1, includes several key phases: dataset input, data preprocessing, building ML and DL models using a stacking-based method, integrating the Kappa Architecture, classifying attacks into binary and multi-class categories, and evaluating the scheme in real-time. These phases ensure that every aspect, from data ingestion to threat mitigation, is seamlessly integrated into a scalable and adaptable security solution.

### 3.1. Dataset Description

The ECU-IoHT dataset, developed by Edith Cowan University (ECU), Australia, is employed in this study to build classification models [25]. As one of the few publicly accessible datasets in the IoMT field, it exposes security vulnerabilities in systems where health monitoring sensors tracking parameters like temperature, blood pressure, and heart rate are connected to networks. The dataset consists of 111,207 records and includes features such as protocol, data flows, timestamps, source and destination bytes, packets, IP addresses, and labels for both normal and attack scenarios. The dataset includes attacks such as ARP spoofing, DoS, Smurf, Port Scan, and normal traffic. These attacks target medical devices, phones, and computers connected through various communication channels. The inclusion of multiple threats combined with medical sensors enhances the dataset’s realism and uniqueness. Cyberattacks are systematically classified into distinct types by analyzing audit records and monitoring network activities.

This dataset is essential for the proposed study as it provides a realistic scenario integrating healthcare devices, networks, and various attack types. The target of binary classification is to detect normal and attack traffic, while multi-class classification aims to detect and classify traffic into ARP spoofing, DoS, Smurf, Port Scan, or no attack. Figure 2 and Figure 3 illustrate the distribution of normal versus attack traffic and the distribution of specific attack types, respectively.

### 3.2. Data Preprocessing

Data cleaning, a crucial step in data preprocessing, ensures data accuracy and reliability by detecting and correcting errors or inconsistencies. Examples of data cleaning techniques applied to the ECU-IoHT dataset include handling missing values and addressing outliers [26]. Missing values were managed through imputation and deletion, while outliers were handled through removal and transformation [27]. These steps collectively enhanced the quality and integrity of the dataset, ensuring its suitability for subsequent analysis.

Since the dataset consists of numerical features/values, a normalization procedure was applied. Normalization is a method used to adjust the values of numerical features so they fall within a specific range, typically between 0 and 1. This ensures that all features have an equal influence on the analysis, regardless of their original scales. Common normalization methods include min–max scaling and Z-score normalization [28].

In this paper, however, the Z-score normalization is used to scale data by removing the mean and scaling to unit variance. The formula is as follows:(1)$$Z=\frac{X-\mu }{\sigma }$$
where

*X* is the original data point.μ is the mean of the data.σ is the standard deviation of the data.

The mean (μ) is calculated as follows:(2)$$\mu =\frac{1}{N}\sum_{i=1}^{N}X_{i}$$

The standard deviation (σ) is calculated as follows:(3)$$\sigma =\sqrt{\frac{1}{N}\sum_{i=1}^{N}{\left(X_{i}-\mu \right)}^{2}}$$

Recall that the ECU-IoHT dataset comprises 111,207 samples, each containing a total of eight features (see Section 3.1). These features include both numerical and categorical data. To ensure compatibility with machine learning algorithms, categorical features are converted into numerical formats using the one-hot encoding strategy, a widely used encoding technique [27,29], as outlined in Algorithm 1.
**Algorithm 1** One-hot encoding algorithm.**Require:** Categorical feature *X* with *n* samples, containing *k* unique categories**Ensure:** Encoded matrix *O* of size n×k
1:Identify the unique categories in *X*: $C=\{c_{1},c_{2},\cdots ,c_{k}\}$2:Initialize an n×k matrix *O* with all elements set to 03:**for** i=1 to *n* **do**4:   Find the category $x_{i}$ of sample *i* in *X*5:   Locate the index *j* of $x_{i}$ in *C*6:   Set O[i][j]=17:**end for**8:**return** Encoded matrix *O*


The main motivation for applying one-hot encoding is to ensure compatibility with machine learning algorithms by transforming categorical data into a numerical format that these algorithms can process effectively. Many machine learning algorithms, such as logistic regression and SVM, require input features to be numerical due to their reliance on mathematical computations (e.g., matrix operations). However, categorical data often lack a natural numerical representation, and assigning arbitrary numerical values can mislead the algorithm by implying an ordinal or continuous relationship between categories where none exists.

One-hot encoding resolves this by representing each category of a categorical variable as a binary vector. Each category is assigned a unique binary feature where one element is “1” (indicating the presence of that category), and all others are “0”. This approach avoids introducing unintended ordinal relationships and ensures that each category is treated as distinct and independent.

In the ECU-IoHT dataset, consider a categorical variable such as “attack type”, which includes categories like “ARP spoofing”, “DoS”, “Smurf”, “Port Scan”, and “No Attack”. Applying one-hot encoding to the variable results in five binary variables, each representing one of these attack types. For instance, if an observation corresponds to an “ARP spoofing” attack, the binary variable for “ARP spoofing” is set to 1, while the variables for “DoS”, “Smurf”, “Port Scan”, and “No Attack” are set to 0. Each binary variable signifies a specific category, and since an observation can belong to only one category at a time, only one binary variable will have the value “1” for each observation. This transformation ensures that machine learning algorithms can process categorical features effectively, avoiding any misinterpretation of relationships between categories, thereby improving the model’s accuracy and robustness for both binary and multi-class classification tasks [30].

Since the ECU-IoHT dataset exhibits class imbalance, data balancing techniques were applied to ensure a fair representation of each class, which is critical for improving the performance of machine learning models. Two popular methods were utilized: oversampling and undersampling. Oversampling involves generating synthetic samples for minority classes using the synthetic minority oversampling technique (SMOTE), which interpolates new samples based on existing ones in the minority class [31]. Conversely, undersampling was employed to randomly reduce the number of samples in the majority classes to balance the dataset. These approaches were combined in a hybrid manner to maintain the dataset’s integrity while mitigating overfitting or information loss.

Given the presence of both numerical and categorical features in the dataset, feature selection strategies were employed to identify the most relevant features for model training. Recursive feature elimination (RFE) [32] was applied to systematically eliminate features with lower importance based on the performance of the machine learning algorithm used. Additionally, statistical methods like mutual information (MI) [33] were used to measure the dependency between features and the target variable, ensuring the selection of features with the highest predictive power. These techniques reduced dimensionality, minimized redundancy, and improved the overall efficiency of the model. It is crucial to partition the data according to feature mapping. The dataset was divided into training and testing subsets using stratification and randomization, ensuring the original class distribution was preserved with an 80:20 split [34].

### 3.3. Stacking-Based Method

Stacking ensemble techniques are integral to the proposed scheme, enabling accurate and reliable predictions by leveraging the combined strengths of multiple models. Unlike single-model approaches, which often struggle with complex data patterns, stacking integrates outputs from diverse base models to construct a meta-model capable of superior performance. This advanced ensemble method enhances prediction accuracy and improves adaptability to varying data characteristics, making it well-suited for the complex requirements of IoMT environments. Studies such as [35,36,37] consistently demonstrate that stacking outperforms individual models by reducing errors and improving generalization capabilities.

In this scheme, stacking combines a variety of machine learning classifiers, such as Naïve Bayes, AdaBoost, and logistic regression (LR), with deep learning models, including DNN, CNN, and LSTM. By capitalizing on the unique strengths of each classifier, this approach significantly enhances the accuracy and robustness of the intrusion detection system. Experimental results confirm that the stacking-based model exceeds the performance of traditional ML and DL methods, achieving superior intrusion detection and classification, as highlighted in the results section. Hybrid models enhance the accuracy and adaptability of IoMT threat detection by integrating machine learning and deep learning techniques. This integration allows for real-time detection of threats while also decreasing the number of false positives.

### 3.4. Kappa Architecture

Jay Kreps introduced the Kappa Architecture as an alternative to the Lambda Architecture, addressing the complexity inherent in managing two separate codebases for batch and real-time processing [38]. The Kappa Architecture offers a streamlined approach to real-time data processing by focusing on scalability, fault tolerance, and efficiency. Unlike the Lambda Architecture, which relies on a dual-layer system, the Kappa Architecture simplifies data management by eliminating the batch layer and integrating batch and streaming capabilities into a single, unified streaming layer. The Kappa Architecture provides a scalable and efficient solution for real-time data stream processing, enabling low-latency threat detection in IoMT systems without overburdening resource-constrained devices.

Within this architecture, frameworks like Apache Spark Streaming enable low-latency operations by processing data as discretized streams (DStreams), ensuring near-real-time data handling. Machine learning models, developed using tools like Spark MLlib, allow for predictive analytics to be conducted on both historical and streaming data. As illustrated in Figure 4, the generated predictions are routed to the service layer, where they are stored in databases such as MySQL, Cassandra, or Hadoop for further utilization in real-time analysis, visualization, and API integration [39].

To enhance malicious traffic detection, the stacking ensemble method is integrated into the processed data framework of the Kappa Architecture. Stacking-based machine learning and deep learning models are trained on historical data and then deployed within the streaming layer. These models provide real-time predictions, which are transmitted to the serving layer for evaluation. The serving layer applies well-known evaluation metrics, as demonstrated in the results section, to derive both binary and multi-class classifications.

The Kappa Architecture’s use of Spark Streaming and ensemble algorithms ensures efficient and accurate real-time processing of traffic data streams. This capability enables the architecture to robustly detect malicious traffic in dynamic environments, highlighting its effectiveness in real-time applications. Our IDS scheme, as illustrated in Figure 5, integrates the IDS in the router within a Kappa Architecture, allowing efficient real-time detection by central monitoring of all network traffic, as shown in the testbed in [25]. This strategic placement ensures rapid detection of both internal and external threats while reducing latency. Kappa’s real-time data pipelines and optimization techniques further enhance performance, allowing the IDS to process traffic within milliseconds. By centralizing detection at the router, this scheme reduces complexity, improves scalability, and eliminates the need for endpoint-specific monitoring. Consequently, it provides secure, low-latency responses, making it highly suitable for critical applications such as IoMT.

### 3.5. Real-Time IDS Configuration Parameters

To ensure efficient real-time intrusion detection, the IDS was meticulously configured to meet the stringent requirements of IoMT environments. Below is a detailed overview of the system’s parameters:

#### Research Environment

The system was developed and tested in a highly optimized computational environment, comprising the following components:**Programming language:** Python 3.10.11 was selected for its extensive library support and seamless integration with machine learning and deep learning frameworks.**Deep and machine learning framework:** PyTorch 2.2.1 offers flexibility for implementing advanced models and ensuring efficient computational performance.**Hardware specifications:** The system was powered by a 12th Gen Intel Core i7-1260P processor with 10 cores and 16 threads, providing robust computational capability for real-time data processing. It included 16 GB of RAM, enabling efficient handling of high-throughput streaming data, and an NVMe SSD, which ensured rapid data access and minimized input/output bottlenecks, thereby optimizing the overall system performance.

This configuration was designed to handle large-scale data streams with minimal latency, ensuring the IDS operates efficiently in real-time scenarios.

## 4. Result and Discussion

In this section, we evaluate the proposed scheme for developing an IDS designed to detect and classify attacks using DL and ML models. The effectiveness and accuracy of the IDS are enhanced through the application of a stacking ensemble method, which combines predictions from multiple models. The integration of the scheme within the Kappa Architecture enables real-time data processing and analytics. The effectiveness and robustness of the proposed IDS are validated using a range of performance metrics, including evaluations for both binary and multi-class classification. By utilizing a streaming layer to process data as they are received, the Kappa Architecture guarantees real-time intrusion detection (ID). This enables immediate utilization of the ML and DL models for attack detection and classification without the need to wait for batch updates. The scheme’s ability to detect attacks with minimal delay is demonstrated, while performance parameters, including accuracy, precision, recall, and F1 score, are assessed on the ECU-IoHT dataset.

### 4.1. Evaluation Metrics

To evaluate our scheme, we utilized well-known evaluation metrics, including accuracy, recall, precision, receiver operating characteristic (ROC), and F1 score. These metrics are delineated in relation to true positives (TPs), false positives (FPs), true negatives (TNs), and false negatives (FNs).

Accuracy is defined as the ratio of instances that are correctly predicted to the total number of instances. Mathematically, it is expressed as follows:(4)$$\mathrm{Accuracy}=\frac{TP+TN}{TP+TN+FP+FN}$$

Precision is defined as the ratio of correctly identified positive instances to the total number of instances classified as positive, including both correctly and incorrectly classified samples. Mathematically, it is expressed as follows:(5)$$\mathrm{Precision}=\frac{TP}{TP+FP}$$

Recall is defined as the proportion of actual positive samples that are correctly identified out of the total number of positive samples. Mathematically, it is expressed as follows:(6)$$\mathrm{Recall}=\frac{TP}{TP+FN}$$

The F1 score, which represents the harmonic mean of precision and recall, typically ranges from 0.0 to 1.0. A higher F1 score indicates a better balance between precision and recall, reflecting improved model performance. Mathematically, it is expressed as follows:(7)$$F1=\frac{2\times \mathrm{Precision}\times \mathrm{Recall}}{\mathrm{Precision}+\mathrm{Recall}}=\frac{2\times TP}{2\times TP+FP+FN}$$

ROC AUC is a crucial metric for evaluating model performance, reflecting improved classification accuracy in both normal and attack scenarios. A high ROC value indicates an effective model. Specificity and sensitivity, which are equivalent to recall, are also considered.(8)$$Specificity=\frac{TN}{FP+TN}$$(9)Sensitivity=Recall

### 4.2. Binary Classification-Based ML Within the Kappa Architecture

By employing techniques such as data balancing, feature selection, and grid search for hyperparameter optimization, we enhanced the predictive capabilities of the models [40]. To ensure a comprehensive evaluation, we applied a five-fold cross-validation technique. As shown in Table 2, the evaluation of ML-based classifiers for binary classification shows that the stacking classifier stands out as the top performer, achieving an accuracy of 0.99, precision of 0.98, recall of 0.99, and an F1 score of 0.99. These results were achieved using the Kappa Architecture, which ensured real-time data streaming and processing, enabling the stacking classifier to effectively detect attacks with minimal latency. Its detection time of 0.0389 s per batch further demonstrates its efficiency, making it the best for real-time malicious traffic detection. AdaBoost also performed well, with an accuracy of 0.98, precision of 0.98, and recall of 0.99, but it fell short of the stacking classifier in terms of detection speed. On the other hand, logistic regression and Naïve Bayes, with accuracies of 0.91 and 0.89, respectively, delivered a lower performance in the precision, recall, and F1 scores, making them less suitable for high-demand real-time scenarios. Overall, the results in Table 1 highlight the clear advantages of ensemble methods like stacking in delivering the highest accuracy, robustness, and real-time efficiency for IoMT applications. Figure 6 visualizes the performance metrics of the evaluated models across four key metrics.

### 4.3. Multi-Class Classification-Based ML Within the Kappa Architecture

Table 3 presents the results of the multi-class classification analysis, where the Kappa Architecture ensures continuous streaming and accurate classification of diverse attack types in real-time, enhancing the scheme’s ability to detect and classify traffic into ARP spoofing, DoS, Smurf, and Port Scan. The performance of ML-based classifiers demonstrates that ensemble methods, particularly stacking, perform exceptionally well in multi-class classification tasks. The stacking classifier has emerged as the most accurate and reliable model, achieving an overall accuracy of 0.9559 and precision, recall, and F1 score values exceeding 0.955. These metrics highlight its robustness and ability to effectively handle diverse multi-class scenarios. The findings emphasize that the stacking classifier provides highly reliable and accurate predictions across all classes by leveraging the strengths of multiple models in an effective approach. AdaBoost performed well, achieving an accuracy of 0.8992, precision of 0.9048, and recall of 0.8992. Although its F1 score of 0.8959 is satisfactory, it fell short of the stacking classifier in all evaluation metrics. Logistic regression performed slightly better than Naïve Bayes, with an accuracy of 0.8983, precision of 0.9004, and recall of 0.8983, yet it still underperformed behind ensemble approaches. Naïve Bayes obtained the lowest metrics, with an accuracy of 0.8514, precision of 0.8804, recall of 0.8514, and an F1 score of 0.8292, highlighting its limitations in handling multi-class classification tasks. These results emphasize the highest performance of ensemble methods, such as stacking, which are the most effective choice for multi-class classification in complex datasets. Their higher accuracy, reliability, and adaptability make them a more robust option compared to traditional models. Figure 7 visualizes the performance metrics of the evaluated models across the key metrics.

Table 4, Table 5, Table 6 and Table 7, respectively, present the experimental results for the models Naïve Bayes (NB), logistic regression (LR), AdaBoost, and the Stacking method, implemented using the Kappa Architecture, in terms of evaluation metrics for each attack class. These tables provide metrics such as precision, recall, F1 score, and ROC AUC, offering a comprehensive comparison of each model’s performance across attacks, including ARP spoofing, DoS, Smurf, and Port Scan. This result highlights how effectively each model detects different types of attacks, showing the strengths and limitations of traditional ML-based classifiers and the stacking method.

Across all models, the highest evaluation metrics were observed in the detection of ARP spoofing. For other attacks, such as DoS, Smurf, and Port Scan, the performance varied, but Stacking and AdaBoost outperformed Naïve Bayes (NB) and logistic regression (LR). The Stacking model outperformed AdaBoost and the other models in all four attacks, obtaining the highest scores in precision, recall, F1 score, and ROC AUC when identifying the best model overall. The stacking classifier’s detection times for each attack were evaluated as they showed the highest performance across all evaluation metrics. With an average frame detection time of 0.0004 s and combined detection times ranging from 28.3010 s Smurf attack to 29.0831 s ARP spoofing, the stacking classifier showed reliable detection times. The ability to maintain real-time detection capabilities while achieving high performance is shown by these results.

### 4.4. Binary Classification-Based DL Within the Kappa Architecture

As shown in Table 8, the evaluations of three enhanced deep learning models—enhanced deep neural network (DNN), long short-term memory (LSTM), and convolutional neural network (CNN1D)—on a binary classification task demonstrates their effectiveness in distinguishing between the two classes. Binary classification based on a DL model highlights the ability of the Kappa Architecture to leverage DL for accurate and timely intrusion detection. The enhanced DNN achieved lower performance, with an accuracy of 96.73%, precision of 96.78%, recall of 96.73%, and an F1 score of 96.65%. However, the slightly lower F1 score compared to accuracy suggests a minor imbalance between false positives and false negatives, indicating potential challenges in certain classifications. This may be attributed to the DNN’s limitations in capturing temporal or spatial patterns present in the dataset. In contrast, the enhanced LSTM and CNN1D showed significantly better performance, both achieving accuracy, precision, recall, and F1 scores exceeding 99%. The LSTM’s metrics, such as 99.11% accuracy and 99.13% precision, highlight its strength in capturing sequential dependencies, making it particularly effective for datasets with temporal patterns. Similarly, CNN1D’s ability to detect local patterns is evident in its 99.10% accuracy and 99.11% F1 score, emphasizing the importance of spatial feature extraction for this dataset. The evaluation of individual models—DNN, LSTM, and CNN1D—along with the stacking ensemble technique, provides valuable insights into their individual and collective performance. By focusing on metrics such as accuracy, precision, recall, and F1 score, the analysis demonstrates how each model addresses binary classification tasks and highlights the advantages of using ensemble stacking for enhanced generalization. The stacking ensemble model combines the predictions of the DNN, LSTM, and CNN1D through a meta-learner. While the base models perform strongly on their own, the stacked model achieves superior results with an accuracy of 0.9913, precision of 0.9915, recall of 0.9910, and an F1 score of 0.9912.

### 4.5. Multi-Class Classification-Based DL Within the Kappa Architecture

In multi-class classification tasks, where balancing precision and recall is critical, the stacking ensemble offers a more reliable solution compared to individual models. Its performance demonstrates the ability to minimize errors such as false positives and false negatives while maintaining high overall accuracy. This is particularly valuable in critical applications like intrusion detection, fraud prevention, and healthcare diagnostics, where the cost of misclassification can be substantial. The evaluation of the three models—DNN, LSTM, and CNN1D—on the given dataset provides significant insights into their strengths and weaknesses. While each model exhibits high overall classification performance, as reflected in their precision, recall, F1 scores, and accuracy metrics, their ability to handle the dataset’s complexity and balance class-specific detections varies, highlighting their individual limitations and potential areas for improvement. The DNN achieved an impressive overall accuracy of 99%, with precision and recall of 99%, and an F1 score of 97%, demonstrating its effectiveness in identifying positive cases and its strong generalizability. These results highlight the suitability of DNN for datasets where majority-class detection is critical. The LSTM slightly outperforms the DNN, achieving an overall accuracy of 99%, with an impeccable precision of 100%, recall of 99%, and an F1 score of 98%. This performance underscores the strength of the LSTM in capturing sequential dependencies, allowing it to effectively model temporal or contextual relationships in the dataset. In contrast, the CNN1D shows slightly lower but still substantial performance, with an accuracy of 96%, precision of 96%, recall of 95%, and an F1 score of 94%. These metrics reflect CNN’s ability to extract spatial features effectively, although it lags behind LSTM in handling sequential data. The stacking ensemble demonstrates the advantages of combining diverse models for better overall performance. Although the LSTM delivered superior individual results, the ensemble model provides more balanced performance across all metrics, ensuring robustness against data variability and edge cases. This is particularly valuable in real-world applications where a single model may not perform consistently well in all scenarios. The stacking approach takes advantage of DNN’s generalization ability, LSTM’s exceptional sequential pattern recognition, and CNN1D’s spatial feature extraction, creating a comprehensive and robust prediction framework, as shown in Table 9. The multi-class classification based on DL models further validates the use of the Kappa Architecture to detect and classify traffic into ARP spoofing, DoS, Smurf, and Port Scan with minimal latency. The stack method achieved high predictive performance with a detection time of 0.8880 milliseconds. This approach involves aggregating the predictions of individual models (DNN, LSTM, and CNN1D) and utilizing a meta-model for final classification.

In Figure 8 and Figure 9, the comparison between models is presented in terms of accuracy, precision, and F1 score for both binary and multi-class classification tasks. It can be observed that the stacking method outperformed the other models in both tasks, demonstrating its effectiveness in detecting and classifying attacks within the IoMT environment.

The integration of the Kappa Architecture significantly reduced detection latency, as shown in Figure 10. By optimizing data pipelines and streamlining execution, Kappa reduced prediction times from seconds to milliseconds. For example, the detection times for LSTM, DNN, and CNN1D were 0.1628 ms, 0.1128 ms, and 0.3051 ms, respectively, while the stacking ensemble achieved 0.8880 ms, highlighting their readiness for real-time applications. In contrast, without Kappa, detection times increased drastically, with stacking requiring 61.3366 s due to inefficiencies in sequential execution and data handling. These findings underscore the critical role of architectures like Kappa in enabling the deployment of DL models in latency-sensitive environments, such as IoMT systems.

The results emphasize the transformative impact of the Kappa Architecture in meeting real-time system requirements efficiently.

Table 10 demonstrates the effectiveness of our scheme, which utilizes a DL-based stacking method, in evaluating metrics such as precision, recall, and F1 score. The results highlight its ability to detect and classify both binary and multi-class scenarios, enhancing IoMT security compared to recently proposed models using the same dataset. A comparison with recently proposed models confirms that the integration of the Kappa Architecture enhances both the performance and real-time capabilities of the proposed IDS.

## 5. Conclusions

IoMT is transforming healthcare by connecting medical devices, sensors, and systems to deliver advanced patient care through remote monitoring, diagnosis, and treatment. However, this technological advancement also brings significant cybersecurity risks due to the sensitive nature of IoMT data and its critical role in patient safety. Cyberattacks targeting IoMT devices can disrupt healthcare services, compromise data integrity, and pose serious threats to patient well-being. Addressing these challenges requires strong and adaptive security solutions that can operate effectively in complex and high-stakes IoMT environments.

This research focuses on developing an IDS specifically designed for IoMT networks. The proposed IDS leverages ML and DL techniques to detect and classify cyber threats with high accuracy. By utilizing a stacking ensemble method, the IDS combines the predictive strengths of multiple ML and DL classifiers, enhancing its ability to detect complex and evolving attack patterns. Additionally, the implementation of the IDS within a Kappa Architecture framework ensures real-time data processing and threat detection, which is critical for IoMT systems where delays in identifying cyber threats could directly impact patient safety.

The IDS demonstrated its effectiveness by accurately detecting and classifying various attack types, including ARP spoofing, DoS, Smurf, and Port Scan. Its performance on key metrics such as accuracy, precision, recall, and F1 score highlights the potential of advanced ML and DL techniques in securing IoMT systems. The integration of ensemble methods, particularly stacking, proved crucial in reducing false positives and improving detection reliability, making the system highly suited for the dynamic and sensitive IoMT environment. The proposed scheme, specifically utilizing the stacking method based on DL, demonstrates high effectiveness, achieving a detection accuracy of 0.991 in binary classification and 0.993 in multi-class classification for detecting and classifying attacks.

However, several limitations must be acknowledged. The reliance on the ECU-IoHT dataset raises concerns about generalizability to other IoMT environments with different network configurations or attack scenarios. While the Kappa Architecture enhances real-time capabilities, its scalability and performance under extreme data loads or diverse attack types remain to be further validated. Additionally, the computational requirements of the stacking ensemble approach may pose challenges for resource-constrained IoMT devices. Despite the IDS’s promising results, the potential for false positives or negatives, along with the complexity of the ensemble model, limits its interpretability and practical deployment in high-stakes healthcare environments. Furthermore, the evolving nature of cyber threats necessitates frequent updates to the IDS, highlighting the need for adaptive and resource-efficient solutions.

Future work will focus on developing adaptive, resource-efficient IDSs using machine learning and deep learning to dynamically detect advanced threats in IoMT devices. 

## Figures and Tables

**Figure 1 sensors-25-00624-f001:**
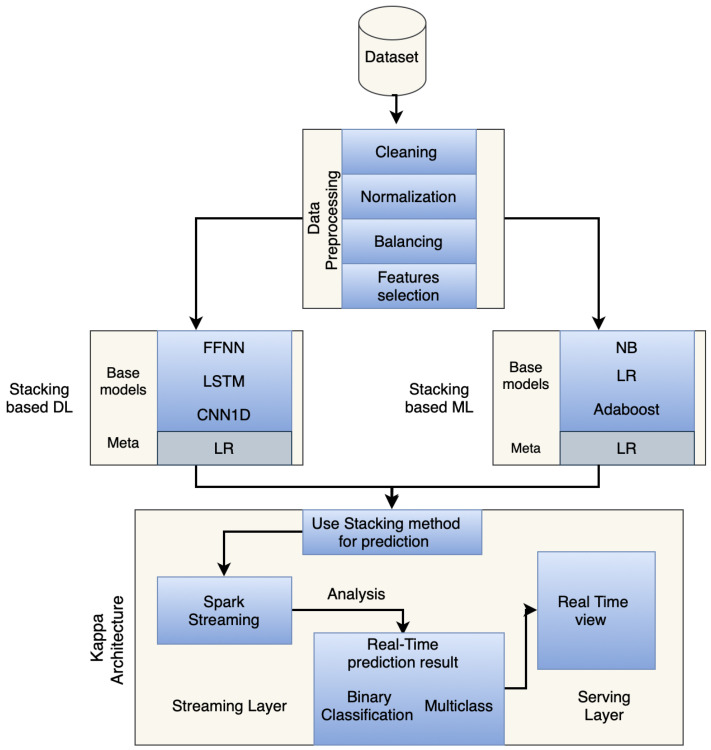
The workflow of the proposed model.

**Figure 2 sensors-25-00624-f002:**
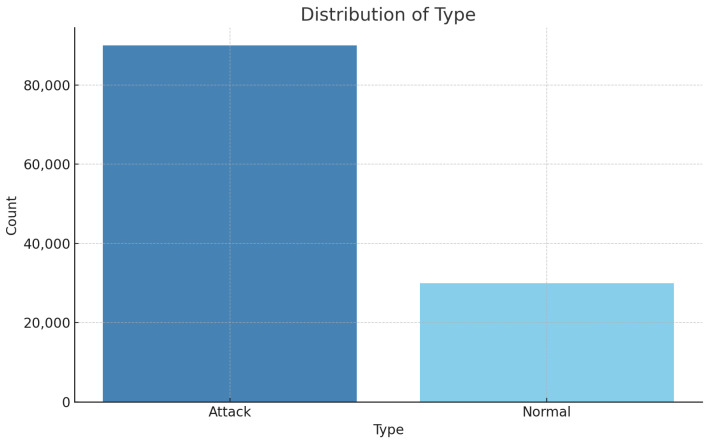
Overview of attack and normal instance distribution.

**Figure 3 sensors-25-00624-f003:**
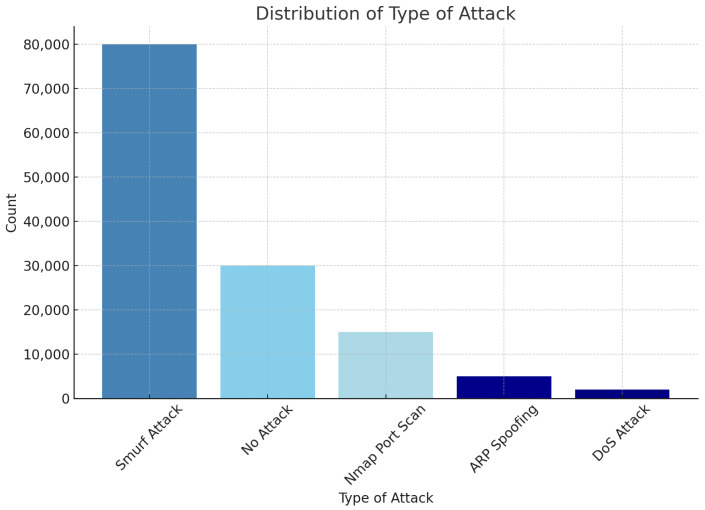
Summary of the dataset’s statistical attributes.

**Figure 4 sensors-25-00624-f004:**
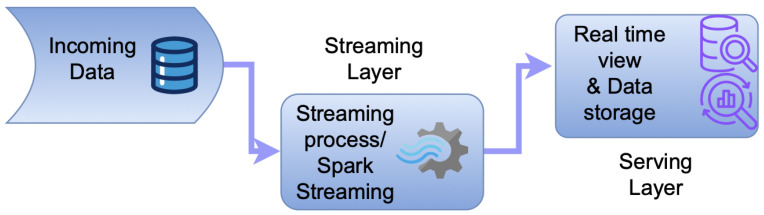
Kappa architecture.

**Figure 5 sensors-25-00624-f005:**
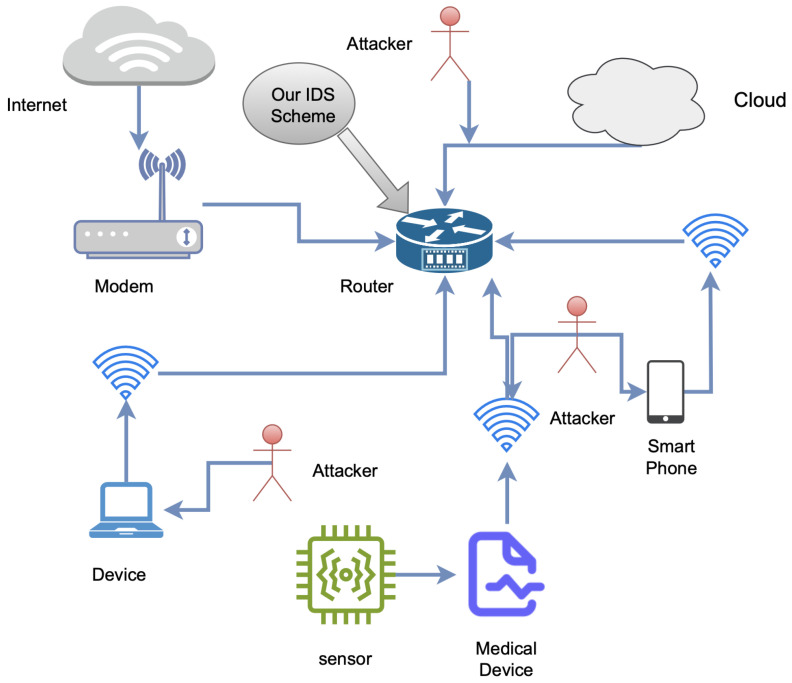
The architecture of the IoMT network with our proposed scheme integrated.

**Figure 6 sensors-25-00624-f006:**
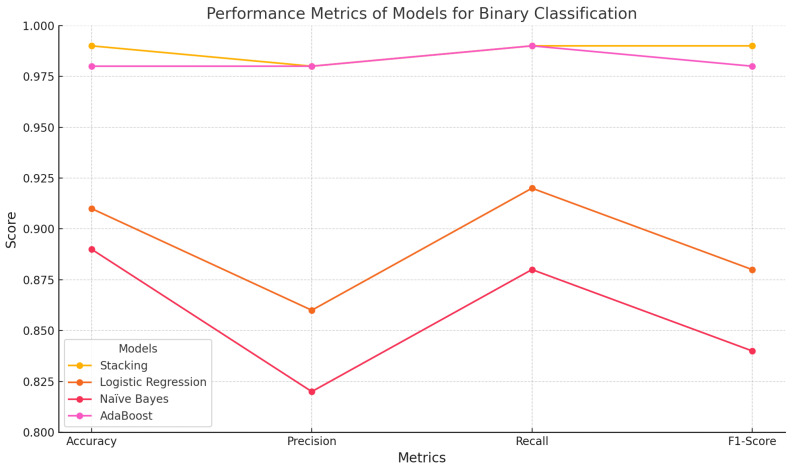
Performance Metrics of Stacking and Machine Learning Models within the Kappa Architecture for Binary Classification.

**Figure 7 sensors-25-00624-f007:**
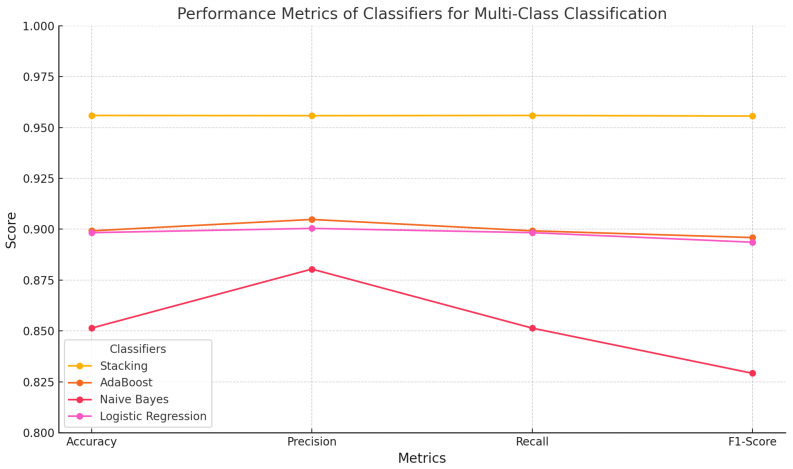
Performance metrics of stacking and machine learning models within the Kappa Architecture for multi-class classification.

**Figure 8 sensors-25-00624-f008:**
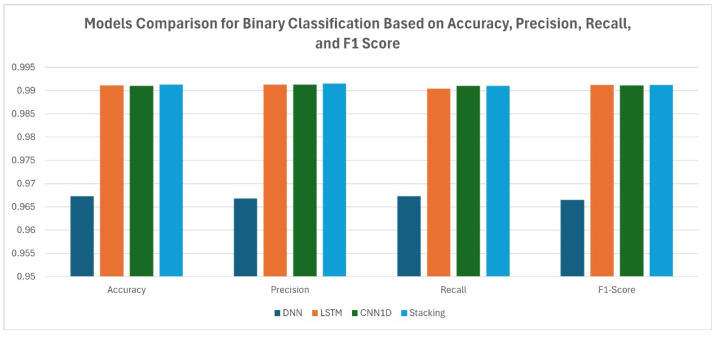
Model comparison for binary classification.

**Figure 9 sensors-25-00624-f009:**
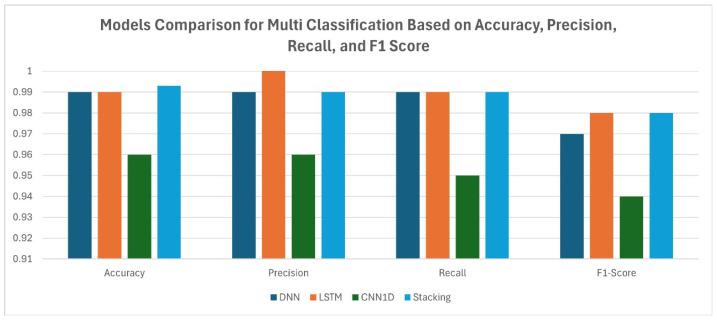
Model comparison for multi-class classification.

**Figure 10 sensors-25-00624-f010:**
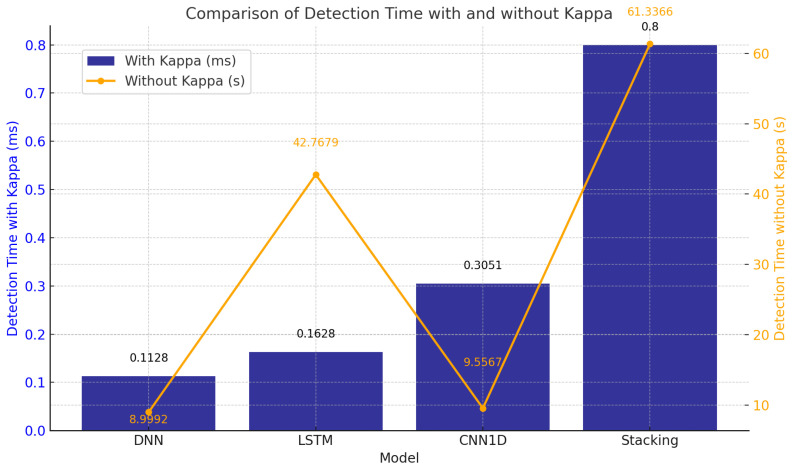
Detection time comparison: with vs. without Kappa Architecture.

**Table 1 sensors-25-00624-t001:** Comparison between our scheme and recently proposed approaches.

Ref.	Year	ML	DL	Ensemble Method	Real-Time Architecture	Binary Classification	Multi-Class	Dataset
[18]	2023	No	Yes	No	No	Yes	No	Physio-Net
[19]	2023	Yes	No	No	No	Yes	No	Heart disease
[7]	2023	Yes	No	Yes	No	Yes	No	TON-IoT
[20]	2023	No	Yes	No	No	Yes	Yes	WUSTL-EHMS
[21]	2023	Yes	Yes	No	No	Yes	No	WUSTL-EHMS 2020, ECU-IoHT
[22]	2024	Yes	Yes	No	No	Yes	No	ECU-IoHT
[23]	2024	No	Yes	No	No	Yes	Yes	ECU-IoHT
[24]	2024	Yes	No	Yes	No	Yes	No	UNSW-NB15
[8]	2024	Yes	No	Yes	No	Yes	No	WUSTL-EHMS
Our work	2024	Yes	Yes	Yes	Yes	Yes	Yes	ECU-IoHT

**Table 2 sensors-25-00624-t002:** Performance metrics of stacking and ML models within the Kappa Architecture for binary classification.

Models	Accuracy	Precision	Recall	F1 score
Stacking	0.99	0.98	0.99	0.99
Logistic regression	0.91	0.86	0.92	0.88
Naïve Bayes	0.89	0.82	0.88	0.84
AdaBoost	0.98	0.98	0.99	0.98

**Table 3 sensors-25-00624-t003:** Performance metrics of stacking and ML models within the Kappa Architecture for multi-class classification.

Classifier	Accuracy	Precision	Recall	F1 score
Stacking classifier	0.9559	0.9558	0.9559	0.9556
AdaBoost	0.8992	0.9048	0.8992	0.8959
Naïve Bayes	0.8514	0.8804	0.8514	0.8292
Logistic regression	0.8983	0.9004	0.8983	0.8936

**Table 4 sensors-25-00624-t004:** Metrics for logistic regression across different attack classes.

Attack Class	Precision	Recall	F1 score	ROC AUC
ARP spoofing	1.0000	1.0000	1.0000	1.0000
DoS attack	0.8722	0.9411	0.9053	0.9818
Nmap Port Scan	0.7885	0.9414	0.8582	0.9844
Smurf attack	0.9835	0.9973	0.9904	0.9992

**Table 5 sensors-25-00624-t005:** Metrics for Naïve Bayes across different attack classes.

Attack Class	Precision	Recall	F1 score	ROC AUC
ARP spoofing	1.0000	1.0000	1.0000	1.0000
DoS attack	0.7408	0.9831	0.8450	0.9853
Nmap Port Scan	0.7129	0.9469	0.8134	0.9658
Smurf attack	1.0000	0.9972	0.9986	0.9997

**Table 6 sensors-25-00624-t006:** Metrics for AdaBoost across different attack classes.

Attack Class	Precision	Recall	F1 score	ROC AUC
ARP spoofing	1.0000	1.0000	1.0000	1.0000
DoS attack	0.7371	0.9364	0.8249	0.9512
Nmap Port Scan	0.9447	0.9514	0.9481	0.9889
Smurf attack	1.0000	0.9973	0.9986	0.9999

**Table 7 sensors-25-00624-t007:** Metrics for the stacking classifier across different attack classes.

Attack Class	Precision	Recall	F1 score	ROC AUC
ARP spoofing	1.0000	1.0000	1.0000	1.0000
DoS attack	0.9207	0.9681	0.9438	0.9920
Nmap Port Scan	0.9456	0.9491	0.9473	0.9874
Smurf attack	1.0000	0.9973	0.9986	0.9999

**Table 8 sensors-25-00624-t008:** Performance comparison of models based on accuracy, precision, recall, and F1 score.

Model	Accuracy	Precision	Recall	F1 score
DNN	0.9673	0.9668	0.9673	0.9665
LSTM	0.9911	0.9913	0.9904	0.9912
CNN1D	0.9910	0.9913	0.9910	0.9911
Stacking	0.9913	0.9915	0.9910	0.9912

**Table 9 sensors-25-00624-t009:** Performance comparison of models for multi-class classification.

Model	Accuracy	Precision	Recall	F1 score
DNN	0.99	0.99	0.99	0.97
LSTM	0.99	1.00	0.99	0.98
CNN1D	0.96	0.96	0.95	0.94
Stacking	0.993	0.99	0.99	0.98

**Table 10 sensors-25-00624-t010:** Evaluation of our stacking-based DL scheme within the Kappa Architecture for binary and multi-class classification in comparison to recent models.

Ref.	Year	Dataset	Model	Accuracy	Precision	Recall	F1
[21]	2023	ECU-IoHT	DNN-FL	0.98	0.93	0.99	-
[22]	2024	ECU-IoHT	EFL-LSTM	0.97	0.93	0.79	0.82
[23]	2024	ECU-IoHT	DAG-LSTM	0.92	0.89	0.92	0.87
Our work	2024	ECU-IoHT	Stacking-DL	0.99	0.99	0.99	0.98

## Data Availability

Data are available upon request.

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
