# Peer review of "Stacking Ensemble Deep Learning for Real-Time Intrusion Detection in IoMT Environments"

_sensors, 2025, doi:10.3390/s25030624_

Round 1

Reviewer 1 Report

Comments and Suggestions for Authors

This research focuses on developing an intrusion detection system powered by machine learning and deep learning techniques specifically designed to protect IoMT devices from cyberattacks. Some issues should be addressed as follows.

1.     In the abstract, the background and motivations of this paper should be simplified, while the novelty and main contributions should be emphasized.

2.     The format of abbreviations should be checked and revised, i.e., “Machine Learning(ML)” should be revised as “Machine Learning (ML)”, and “Internet of Medical Things (IoMT)” has appeared more than once in the main content.

3.     As the background of this paper includes sensors, Internet of Things and deep learning, recent high quality works on the relevant research areas should be introduced to broaden the background, such as Distributed DDPG-based resource allocation for age of information minimization in mobile wireless-powered Internet of Things, IEEE IoTJ.

4.     In the introduction, the contributions of few related works are introduced in detail, and compared with the contributions of this paper.

5.     In the related works, the logic of introducing the contributions of references should be improved, and some references should be updated by recent works on prestigious journals and conferences.

6.     Some paragraphs, such as the first paragraph in Section 3, are too long, and are suggested to be divided into several paragraphs to improve the logic.

7.     Figures should be revised for readability; such as figures 2-3.

8.     Why authors consider the one-hot encoding algorithm to ensure compatibility with machine learning algorithms? Corresponding reasons and justifications should be provided.

9.     Some writing issues should be checked and revised, such as “In [27–29] demonstrate that stacking strategy outperform single models in terms of accuracy and adaptability.”

10.   Evidence should be provided for some statements, such as “Jay Kreps introduced the Kappa Architecture as an alternative to the Lambda Architecture”.

11.   Competitive comparison schemes should be provided in simulations for validation.

Author Response

Thank you for taking the time to provide your feedback. I truly appreciate your insights.

Please find the attached PDF file

Reviewer 2 Report

Comments and Suggestions for Authors

This paper focuses on developing an Intrusion Detection System (IDS) tailored to the Internet of Medical Things (IoMT), a specialized branch of the Internet of Things (IoT) that connects medical devices, sensors, and healthcare systems for remote monitoring, diagnosis, and treatment. Key challenges in IoMT include stringent data security requirements, vulnerability to cyberattacks, and the critical nature of healthcare services. I would suggest revisions as listed below:

1.       How can healthcare providers effectively incorporate IDS systems into existing IoMT infrastructures without disrupting patient care workflows?

2.       Some references about sensor and data-driven approaches to enhance the literature review: Efficient deep data assimilation with sparse observations and time-varying sensors; Reactor field reconstruction from sparse and movable sensors using Voronoi tessellation-assisted convolutional neural networks

3.       How can hybrid models (e.g., combining machine learning with rule-based systems) enhance the detection and prevention of IoMT-specific threats?

4.       What are the implications of resource-constrained IoMT devices (e.g., battery-powered sensors) on the feasibility of deploying this IDS at scale?

Author Response

(The authors gave the same response as above.)

Reviewer 3 Report

Comments and Suggestions for Authors

Paper Overview: This research develops an Intrusion Detection System (IDS) tailored for IoMT environments, leveraging Machine Learning (ML) and Deep Learning (DL) techniques. Traditional IDS solutions often struggle in IoMT settings due to their unique data complexity, regulatory requirements, and high-reliability requirements. The proposed IDS employs a stacking ensemble method, combining multiple ML and DL models to detect and classify cyber threats. By leveraging the strengths of diverse classifiers, the system improves threat detection accuracy and reliability. Implemented within a Kappa Architecture framework for real-time performance, the IDS achieves outstanding results, with detection accuracies of 0.991 for binary classification and 0.993 for multi-class classification, demonstrating its effectiveness in protecting IoMT devices from cyberattacks.

My Comments:

The paper is almost a revised copy of the author’s previous publication, “Enhancing Cybersecurity in Healthcare: Evaluating Ensemble Learning Models for Intrusion Detection in the Internet of Medical Things.” The only difference is in the dataset. I don’t find any novelty in this paper. Only changing a dataset with the same old techniques couldn’t contribute to the research community.

Read the following papers; and see the similarities in methodology and result discussion.

Alotaibi, Y.; Ilyas, M. Ensemble-learning framework for intrusion detection to enhance internet of things’ devices security. Sensors 496 2023, 23, 5568. 497 7.

 Alsolami, T.; Alsharif, B.; Ilyas, M. Enhancing Cybersecurity in Healthcare: Evaluating Ensemble Learning Models for Intrusion 498 Detection in the Internet of Medical Things. Sensors 2024, 24

What are the limitations of the proposed system in “Stacking Ensemble Deep Learning for Real-Time Intrusion Detection in IoMT Environment”, and how does this work advance the previous work (papers given above)?

The comparison in Table 10 is misleading. Are all other techniques implemented on the same dataset?

How is Kappa Architecture used for real-time ID? No parameters and performance metrics are discussed for real-time Intrusion Detection (ID) on Kappa Architecture.

In the paper, only the authors used the architecture name, but the results were used to implement and compare existing ML/DL models (without customization) and produce results. 

Comments on the Quality of English Language

English is of readable quality.

Author Response

(The authors gave the same response as above.)

Round 2

Reviewer 1 Report

Comments and Suggestions for Authors

No further comments.

Author Response

Thank you very much

Reviewer 2 Report

Comments and Suggestions for Authors

the paper has been improved

Author Response

Thank you very much 

Reviewer 3 Report

Comments and Suggestions for Authors

I am still not entirely convinced by the modifications and the author's response. I asked the authors about the novelty of this article compared to the author's previous two articles. However, they did not respond to my question.

Comments on the Quality of English Language

English is fine.

Author Response

Thank you for taking the time to provide your feedback. I appreciate your insights.

Please find the attached PDF file
